# The Weight of Migration: Reconsidering Health Selection and Return Migration among Mexicans

**DOI:** 10.3390/ijerph182212136

**Published:** 2021-11-19

**Authors:** Aresha M. Martinez-Cardoso, Arline T. Geronimus

**Affiliations:** 1Department of Public Health Sciences, University of Chicago, Chicago, IL 60637, USA; 2Department of Health Behavior and Health Education, University of Michigan, Ann Arbor, MI 48109, USA; arline@umich.edu

**Keywords:** Mexican, Hispanic/Latino paradox, stress, migration

## Abstract

While migration plays a key role in shaping the health of Mexican migrants in the US and those in Mexico, contemporary Mexican migration trends may challenge the health selection and return migration hypotheses, two prevailing assumptions of how migration shapes health. Using data from the Mexican Family Life Survey (2002; 2005), we tested these two hypotheses by comparing the cardiometabolic health profiles of (1) Mexico–US future migrants and nonmigrants and (2) Mexico–US return migrants and nonmigrants. First, we found limited evidence for health selection: the cardiometabolic health of Mexico–US future migrants was not measurably better than the health of their compatriots who did not migrate, although migrants differed demographically from nonmigrants. However, return migrants had higher levels of adiposity compared to those who stayed in Mexico throughout their lives; time spent in the US was also associated with obesity and elevated waist circumference. Differences in physical activity and smoking behavior did not mediate these associations. Our findings suggest positive health selection might not drive the favorable health profiles among recent cohorts of Mexican immigrants in the US. However, the adverse health of return migrants with respect to that of nonmigrants underscores the importance of considering the lived experience of Mexican migrants in the US as an important determinant of their health.

## 1. Introduction 

Among people of Mexican descent in the United States (US), nativity and generational status are consistently associated with adiposity, diabetes, and cardiometabolic health. In general, recent Mexican immigrants present with lower levels of cardiometabolic risk factors and adverse outcomes in comparison to their US-born Mexican American counterparts and, in some cases, non-Latino US-born Whites [1]. For immigrants, more time in the US is also associated with increasingly deleterious health outcomes [2,3], namely, obesity, elevated blood pressure, and type-2 diabetes. Advantageous cardiometabolic health among recent Mexican immigrants is one example of what has been coined the Latino/Hispanic or immigrant health paradox, where despite their considerable social and economic vulnerabilities, immigrants have lower rates of mortality, adverse birth outcomes, and cardiovascular mortality risk than US-born Mexican Americans [4,5,6].

In this research literature, health selection, return migration, and acculturation hypotheses are proposed to explain the disparate health outcomes between Mexican immigrants and US-born Mexicans and between recent and long-stay Mexican immigrants. Health selection arguments suggest that Mexican migration is positively selected on the basis of health and other sociodemographic characteristics associated with health, leading the healthiest individuals to engage in migration [7]. In addition, the return migration/salmon bias hypothesis proposes that immigrants who become unhealthy in the US might return to their country of origin for care, leaving the healthiest immigrants in the US [8]. Therefore, research that compares the health of immigrants and that of US-born may be mis-specified because it fails to account for these in-migration and out-migration selection processes and compares an artificially healthy cohort of immigrants to the US-born population with a more diverse health profile [9].

Moreover, researchers have proposed that acculturation towards the US unhealthy behaviors and social practices might also drive the health disadvantages of US-born and long-stay migrants. Some argue, for example, that health behaviors that are associated with cardiovascular disease may be better in sending countries compared to the US, with more individuals engaging in behaviors that promote health, such as physical activity and healthy dietary practices [10]. Therefore, immigrants may arrive in the US and still engage in cardiometabolic-related behaviors that promote health, such as physical activity and healthy dietary practices [11]. In the US, recent immigrants may also retain positive social networks that promote their health [11]. With more time in the US, however, acculturation arguments suggest that immigrants shift to US norms, which worsens the health behaviors, social, and cultural practices that were protective of immigrants’ health [12].

The evidence supporting these hypotheses is mixed, calling into question these potential drivers of immigrant health. For example, whether health selection plays a large role in the calculus of migration as opposed to other drivers such as economic instability and violence is subject to debate [13]. In addition, new evidence demonstrates that the health behaviors of recent immigrants may not, in fact, be better than those of US-born or long-stay migrants [14]. Similarly, the role of return migration due to health may be overestimated, since return migration to Mexico is often due to both involuntary and voluntary factors including family reunification, employment prospects, and deportation [15]. Instead, drawing on scholarship in migration, racialization, and health, a new stream of research argues that the deleterious health of migrants may be linked to chronic exposure to social, political, and environmental stressors over time and across generations [16]. These more contemporary arguments have offered an alternative explanation for the declines in immigrants’ health with longer residence in the US, yet traditional health selection and acculturation explanations are replete in the literature.

New methodological approaches also offer innovative ways to evaluate the hypotheses of the immigrant health advantage. Most recently, research has shifted from cross-sectional data in the US to the use of binational data from both sending and receiving countries [17,18,19,20,21,22]. This approach has enabled comparisons between behaviors and outcomes of migrants and their compatriots who do not migrate, an arguably more appropriate comparison group to assess the possibility of health selection and health-driven return migration.

Our analysis contributes to these debates by using data from the Mexican Family and Life Survey (MxFLS) to explore health selection and return migration among Mexican migrants to the US, with a focus on adiposity, blood pressure, cardiovascular disease, and diabetes. The MxFLS is a recent nationally representative study that measures health and migration in the Mexican population considering future and return migration. As such, our analysis lends new evidence about contemporary migration and health. In addition, we advance the literature findings by using health data that were collected by trained staff, rather than relying only on self-rated health measures, as most surveys do.

First, within the population of Mexico, we compared cardiometabolic health between Mexico–US future migrants (*n* = 322) and nonmigrants (*n* = 14,441). We hypothesized that Mexicans who migrated to the US between the waves of the MxFLS would have similar cardiometabolic health profiles as Mexicans who did not migrate, adjusting for demographic characteristics. Next, we compared the cardiometabolic health profiles of Mexico–US return migrants (*n* = 276) and those in Mexico who never out-migrated (stayers, *n* = 14,441). We hypothesized that return migrants would have worse cardiometabolic health than stayers. Furthermore, we explored associations between health and time spent in the US, age at migration, and documentation status to test whether these dimensions of migration to the US were associated with poorer health. Finally, we tested whether health behaviors related to physical activity or smoking mediated the relationship between return migration and health.

## 2. Methods

### Data

We use two waves of data from the Mexican Family Life Survey (MxFLS), a nationally representative and longitudinal survey of households in Mexico [23]. The baseline survey in 2002 collected sociodemographic and anthropometric data on 8440 households and 19,809 adults across 150 communities in Mexico. Private dwellings formed the primary sampling units of this survey, and the sample was designed to be nationally, urban-rurally, and regionally representative of the Mexican population [24]. Sampling units with similar geographic and socio-economic characteristics were grouped into a single stratum, and households were selected among the strata. All adults and children in the sampled households were eligible for inclusion in the MxFLS. The second wave of data was obtained in 2005–2006 by successfully recontacting over 90% of the original household sample, including those who migrated within Mexico or emigrated to the US. We leverage the unique migration and migration history measures in the data to create indicators of future migration to the US and return migration from the US (elaborated below). We limited our sample to adult respondents with measures of their migration history, health, and sociodemographic variables of interest (*n* = 14,763).

## 3. Measures

### 3.1. Migration Indicators

The MxFLS includes a variety of questions about local and international migration, which we leveraged for our analysis. The MxFLS tracked individuals across wave 1 in 2002 and wave 1 in 2005, identifying individuals who lived in Mexico in wave I but moved to the US in wave 2. For the first research question testing health selection, we use this variable to classify *future migrants*, respondents who migrated to the US between waves 1 in 2002 and wave 2 in 2005, and *nonmigrants*, respondents who remained in Mexico during both waves of the study. In total, we identified 322 future migrants and 14,441 nonmigrants.

For the second research question that tested the association between return migration from the US and health, we leveraged a series of variables that asked the participants to detail all the places they had moved to both within and outside Mexico since the age of 12. Using this variable, we identified *return migrants*, respondents who migrated to the US and returned to Mexico, and *stayers*, respondents who never migrated to the US. We classified respondents as return migrants if they reported living in the US for a period of 12 months or longer (*n* = 276). Stayers included respondents who reported that they had never lived in the US (*n* = 14,487). In addition, for the return migrant group, we computed the total number of years spent in the US and age at migration. Finally, we created a variable for documentation status during migration to the US, classifying individuals as undocumented or documented at migration. For those with multiple migration trips to the US, information from the last migration trip was used for the age at migration and documentation status variable.

### 3.2. Cardiometabolic Health Indicators and Health Behaviors

A trained survey staff collected anthropometric health data of the respondents at baseline. Using these data, we computed measures for waist circumference and mean arterial blood pressure (MAP) (SBP + 3 *DBP/3). For sensitivity analysis, we also created measures of body mass index (BMI) (weight (kg)/height(m)^2^) and hypertension. We created dichotomous indicators of elevated levels for each of these measures (elevated waist circumference, obesity, elevated MAP, and elevated blood pressure) using appropriate clinical cutoffs [25,26]. In addition, we created dichotomous measures of diabetes status and cardiometabolic disease (history of heart disease, heart attack, cholesterol/arteriosclerosis, or stroke) based on self-reported data.

Health behaviors included physical activity and smoking status. Using information on the frequency and duration of physical activity, we created a dichotomous indicator of whether the respondents met the criteria for the recommended amount of physical activity (>150 min/week). The respondents were also classified as current, former, or nonsmokers based on a series of questions about their smoking history.

### 3.3. Controls

We controlled for several demographic variables in our models that are associated with the selected health outcomes, including age, gender, marital status, current employment status, highest level of education completed, health insurance status, and household assets. Finally, we included controls for urban/rural residence, since previous research suggests that residents of urban/rural regions have distinct migration patterns.

## 4. Analysis

The analysis was completed using Stata 15. The datasets generated during the current study are available from the corresponding author on reasonable request. Determination of exempt status was obtained from the University of Michigan and University of Chicago BSD/UCMC Institutional Review Boards. Descriptive statistics were generated to assess the quality of the data and the proportion of missing cases. Cases that were missing data on variables of interest were dropped from the analysis. The final analytical sample included 14,763 respondents. Pearson’s chi-square tests and t-tests were used to compare demographic and health characteristics between future migrants and non-migrants as well as between return migrants and stayers at Wave 1.

Next, mixed-effects models were used to examine the association between health at baseline and future migration and between return migration and health. Mixed-effects models were used because data in the MxFLS are clustered on two levels: respondents were nested within families/households, and families/households were grouped within communities. Mixed-effects models allow us to make appropriate statistical inferences while accounting for the multilevel nature of the data. Odds ratios (OR) and 95% CI are reported for dependent variables that were modeled using mixed-effects logistic regression, while beta coefficients are reported for dependent variables that were modeled using mixed-effects linear regression.

First, we tested whether health at wave 1 was associated with Mexico–US migration at wave 2. In these models, Mexico–US migration was the outcome variable, while health variables at baseline were modeled as covariate variables. That is, we tested whether each of the health indicators significantly predicted whether a respondent was a *future migrant* or a *nonmigrant*. To account for potential collinearity among the health variables, each health variable was entered by itself in separate models.

Second, we tested whether return migration from the US was associated with each of the health variables at wave 1. In these models, each health indicator at wave 1 was the outcome variable, and US–Mexico return migration was modeled as a covariate. We then ran a similar series of models with time in the US as a continuous independent variable; non-migrants were assigned a value of 0, while return migrants were assigned a value corresponding to the year(s) they spent in the US. We also modeled time in the US as a categorical variable based on quintile cutpoints to gauge which level of time in the US was most consequential for health. We also tested if associations between return migration or time in the US and health were mediated by smoking and physical activity behaviors by entering these variables in the model and comparing point estimates and *p*-values. Finally, we ran a subset of models among return migrants to test the association between age at migration, documentation status at migration, and health.

## 5. Results

### 5.1. Descriptive Statistics

Table 1, column 1 presents descriptive statistics for the overall sample (*n* = 14,763). There was a larger proportion of women (55%) than of men, and the mean age was 40 years (sd = 16.6). The majority of the respondents were married (67%). In addition, over 60% of the sample had achieved at least a primary school education in Mexico, and half of the respondents (58%) reported that they had worked in the past month.

In terms of health, nearly a quarter of the respondents had an elevated waist circumference, and 26% of the sample was classified as obese, which is consistent with national estimates [27]. Based on blood pressure measures, 38% had an elevated mean arterial pressure, and 37% were classified as having high blood pressure. The prevalence of self-reported diabetes and cardiometabolic disease was 6% and 3%, respectively. Finally, 14% of the respondents were current smokers, and 14% performed at least 150 min of physical activity per week.

### 5.2. Migrant Groups Bivariate Analysis

Next, we compared the sociodemographic and health characteristics of future migrants vs. those of non-migrants (Table 1, Column 2 and 3) and of return migrants vs. those of stayers (Table 1, column 3 and 4). Mexico–US future migrants were respondents who lived in Mexico during Wave 1 of the study in 2002, then lived in the US during Wave 2 in 2005–2006; non-migrants were respondents who remained in Mexico across both waves of the study. The sample comprised a total of 322 Mexico–US future migrants and 14,441 non-migrants. As compared to non-migrants, future migrants were younger and more likely to be men, unmarried, and uninsured. Future migrants had higher rates of primary school completion and were more likely to reside in rural regions of Mexico. In addition, future migrants had lower levels of elevated waist circumference, obesity, high blood pressure, and self-reported diabetes and were more likely to engage in at least 150 min of physical activity per week.

Finally, 276 individuals in the sample were Mexico–US return migrants—individuals who had previously migrated to the US for a period of at least 12 months but had returned to Mexico—whereas 14,487 were stayers, i.e., individuals who had only lived in Mexico. On average, return migrants reported living in the US for 4 years and migrating at the age of 24 years; 75% reported that they were undocumented when they migrated. Compared to stayers, return migrants included a larger proportion of men and were more likely to be married, currently working, uninsured, current smokers, and meet exercise recommendations. Return migrants were largely comparable to stayers across all health variables, based on the bivariate analysis.

### 5.3. Health Selection

We estimated mixed-effects regression models that tested whether health at wave 1 was associated with migration to the US by wave 2. All models included the sociodemographic control variables. We entered each of the health indicators as an independent predictor of future migration without any other health variable to avoid multicollinearity issues. A summary of regression coefficients for each health variable, estimated from separate models, is shown in Table 2; full models with control variables are provided in Appendix A. When entered singly, only elevated waist circumference was significantly associated with migration to the US in wave 2 (*p* < 0.10).

Notably, however, the point estimates for the ORs suggested the possibility of other differences, and the confidence intervals were wide, suggesting imprecise estimates that we attributed to the small sample size of future migrants. Examining the estimated ORs at face value only, they were not in a consistent direction. For example, the estimated ORs suggested that future migrants were more likely to be smokers and to report a history of cardiometabolic disease, yet also more likely to be physically active and have a smaller waist circumference. In other words, we found no consistent evidence for positive health selection, net of controls.

For robustness checks, we tested the models in Table 2 with continuous versions of health variables as well as various comparison groups. In addition, we substituted BMI and hypertension for weight circumference and mean arterial pressure, respectively. The results were consistent. Because future migration was associated with a previous migration to the US and we hypothesized that return migration was independently associated with health, we also controlled for return migration status in our models. We checked whether excluding return migrants from the sample changed these findings and found that the results remained consistent.

### 5.4. Return Migration

Next, we explored whether ever migrating to the US was associated with each of the health variables during wave 1 and modeled the relationship between time in the US and health. In these models, each of the health indicators was separately modeled as the dependent variable, with the return migration indicator included as an independent variable. A summary of the regression coefficients for return migration and time in the US regarding each of the health variables is shown in Table 3.

Return migration was significantly associated with waist circumference and obesity. Return migrants had 41% increased odds of being obese. Waist circumference was also statistically associated with time spent in the US, suggesting a stress-related distribution of adiposity toward the abdomen among return migrants. On average, return migrants had a waist circumference that was 1.49 cm greater than stayers, and each additional year in the US increased waist circumference by a quarter of a centimeter. The inclusion of physical activity or smoking in the models did not significantly change these findings. When we modeled time in the US based on categorical cutpoints, we found that immigrants who spent 2.16–5 years had a waist circumference that was 3.39 cm greater on average than non-migrants, the largest difference in waist circumference across the time in the US groups (Appendix A). Those who spent less than 1.16 years in the US were indistinguishable from non-migrants.

However, we found no significant association between return migration and hypertension, mean arterial pressures, or diabetes. Time in the US was also not associated with high blood pressure or self-reported diabetes. Finally, we tested return migration and time in the US on self-reported cardiometabolic disease. In these models, return migration was not significantly associated with cardiometabolic disease; however, time in the US was associated with increased odds of reporting heart disease, stroke, or atherosclerosis (OR = 1.06, 95% CI 1.003–1.126).

In a sub-analysis among return migrants, we explored whether age at migration and documentation status during migration were associated with the health indicators. Neither was. Given the small sample sizes, however, these point estimates from the models proved to be unstable, with large confidence intervals. Therefore, we hesitate to meaningfully interpret these results.

## 6. Discussion

This analysis drew on a large and multi-thematic dataset of adults in Mexico to understand how health might shape contemporary migration to the US. We also explored how migration to the US might shape health and health behaviors among Mexican migrants who return to their country of origin. In general, we found that a variety of health behaviors, cardiometabolic health risk factors, and cardiometabolic disease were not associated with future migration to the US. Instead, more traditional factors such as age and gender were consistently associated with migration. In addition, having health insurance also reduced the odds of migration. This finding may be due, in part, to the types of employment sectors that provided health insurance to Mexican residents in 2002 (private business and government employees) and to the employment and economic opportunities these individuals had in Mexico compared to uninsured individuals outside of these sectors. We found little and mixed evidence in support of the view that those Mexicans who migrate to the US are positively health-selected. Based on these findings, health was not an important predictor of the respondents’ decision to migrate to the US between the waves of the MxFLS. While there is some evidence of selection in migration by demographic factors, a previous analysis with older Mexican cohorts still found evidence of health selection and health advantages among immigrants, net of these demographic controls. Our analysis departs from these findings. As such, it is unlikely that health selection with respect to cardiometabolic health acts as a major driver of the advantageous health profile of recent Mexican migrants to the US These findings are consistent with those of a similar analysis that explored other health outcomes that may be linked to health selection [28].

While we found that migrants and nonmigrants had similar health profiles before migration, health outcomes among return migrants as compared to those of stayers revealed a different story. Return migrants, those who had ever spent a year or longer in the US, were more likely to be overweight and had a larger waist circumference as compared to stayers. Moreover, return migrants who had lived in the US for more time also had higher waist circumferences and self-reported cardiometabolic disease. However, neither return migration nor time in the US were associated with other measured cardiometabolic indicators including hypertension and diabetes. Our mixed findings may be due to the fact that the health outcomes associated with adiposity take a shorter time to manifest as compared to diabetes, hypertension, and cardiometabolic disease. In a similar analysis with a more restricted sample of male Mexican migrants, Ullman et al. also found return migrants had higher levels of obesity, with no differences in hypertension or diabetes [22]. Similarly an analysis of children in the MxFLS found that children in Mexico within migrant households also had higher levels of obesity [29]. If followed for more time, we might expect that return migrants may in fact display more adverse chronic disease profiles than stayers.

In summary, we found that while the health of Mexico–US migrants was on par with that of their compatriots who did not migrate, the health of US–Mexico return migrants was worse than that of stayers on some indicators. These results have multiple potential explanations. The acculturation hypothesis suggests that return migrants could have adopted worse health behaviors in the US that drove these differences. However, we tested whether physical activity and smoking mediated these associations and found limited support for an assimilation explanation. Other health behaviors, such as diet, were not measured in the study and may have played a role in these outcomes. However, several studies have called into question the acculturation argument, showing that health behaviors among migrants are already poor upon arriving to the US, and few change meaningfully enough to drive health differences (an important exception being smoking) [30,31]. Return migrants in our sample could also have returned to Mexico because they were sick or unhealthy, as suggested by the salmon bias hypothesis. However, scholars speculate that return migration due to health is for serious illnesses such as cancer or disability, rather than the health indicators that emerged in the analysis, i.e., obesity and elevated waist circumference.

Instead, our findings lend support to the notion that the worse health of returning Mexican migrants compared to those who remain in Mexico may reflect the embodiment of the stressful social environment Mexican migrants endure in the US [16,32]. Migrants experience an incredible amount of stress in the process of migrating and adapting to a new place [33]. Mexican migrants, in particular, contend with being defined as marginalized and racialized immigrants in the US and the explicit and implicit experiences of discrimination and othering as a result [34]. These chronic exposures to stress due to the migration experience could over-activate stress responses in the body, which have been linked to adiposity [35,36]. In particular, chronic stress has been specifically indicated in the progression of central adiposity, including waist circumference and visceral fat deposits [37,38]. Our findings that return migration from the US and more time in the US were associated with higher waist circumference point to the potential of this stress exposure hypothesis, warranting future research in this area. For policy and practice, our findings underscore the importance of developing strategies to protect the health of Mexican immigrants beyond traditional health behavior interventions. For example, Mexican consulates have developed *Ventanillas de Salud* programs as one approach to address immigrants social and health vulnerabilities in the US [39]. At a population level, local and state-level immigration policies have also proven to be an important lever to protect immigrants’ health [40]. In addition, given the growing population of return migrants in Mexico, health practitioners in Mexico should monitor the health of Mexicans with previous migration histories in the US as at a potential risk for poor health.

We interpret our findings and their contributions while also noting some important limitations of our analysis and data. The cross-sectional nature of the return migration data limits our ability to infer causation between return migration and health. Those who were return migrants could have had higher levels of adiposity before they migrated to the US as compared to stayers. On average, however, those who were future migrants had smaller waist circumferences than stayers, although it is unclear if these findings are consistent for earlier immigrant cohorts. In addition, given the nature of the data, we were unable to compare the health of return migrants with that of Mexicans who remained in the US; the inclusion of this group in future analysis would help strengthen our findings about the association between migration to the US and adverse levels of adiposity. Looking only at Mexican immigrants who remained in the US, Kaestner et al. found that the length of time in the US was associated with a higher allostatic load, an indicator of stress-mediated wear and tear, net of age, diet, insurance, and other socioeconomic and health behaviors [32]. Our findings related to increased adiposity among return migrants are consistent with such findings and their implications. While our analysis augments the current literature by comparing the health of migrants to that of those who remained in Mexico, future studies that explicitly compare Mexican non-migrants, return migrants, and migrants in the US with larger samples would lend more clarity to the mechanisms driving health differences among these groups. Finally, the age of our data is one drawback, as migration mechanisms and cohorts may have changed between now and the data collection period; yet, the MxFLS is among the most high-quality nationally representative datasets that measure both health and migration in the Mexican population.

## 7. Conclusions

The Mexico–US migration flow represents one of the largest global migration flows, and Mexican migrants account for the largest immigrant-origin group in the US [41]. Furthermore, Mexican migrants comprise 35% of the Mexican-origin population in the US and 20% of the overall Latino population [42,43]. This analysis therefore elucidates some of the potential mechanisms and drivers of Latino health in the US and provides an important examination of the Latino health paradox using data on Mexicans. While the current literature is mixed on how health might shape individuals’ propensity to migrate, how the US shapes migrants’ health, and the various mechanisms that influence these processes, we demonstrated that health does not appear to be a major selection factor for future migration among Mexican adults. However, upon returning to Mexico, adults with migration histories in the US fare worse in weight, specifically, in having a stress-related distribution of adiposity, especially if they lived in the US long. Mexicans’ experience of being “othered” in the US, arguably, has become more severe in light of contemporary efforts toward criminalizing immigration. Future work should explore and examine the unique social and health environments faced by Mexican migrants to the US in light of these findings. More work is also needed to understand how reintegration into one’s country of origin shapes the health of Mexican return migrants.

## Figures and Tables

**Table 1 ijerph-18-12136-t001:** Descriptive Statistics and Bivariate Analysis of Future Migrants vs. Non-migrants and Return Migrants vs. Stayers, MxFLS, Wave 1, *n* = 14,763.

Variable	Total Sample (*n* = 14,763)	Future Migrants (*n* = 322)	Non-Migrants (*n* = 14,441)	Return Migrants (*n* = 276)	Stayers (*n* = 14,487)
	Mean (sd)/%	Mean (sd)/%		Mean (sd)/%	
Age	40(16.6)	29(11.5) ^†^	41(16.6)	39(14.5)	40(16.7)
Female	55	47 ^†^	56	29 ^†^	56
Married	67	51 ^†^	68	75 ^†^	68
Primary School Education	63	76 ^†^	63	67 *	63
Currently Working	57	60	57	68 ^†^	57
Health Insurance	45	21 ^†^	46	36 ^†^	46
Household Assets (ref = owns house)	85	86	84	84	85
Urban Region	57	42 ^†^	58	58	57
Return Migrant	2	9 ^†^	2	--	--
Future Migrant	2	--	--	--	--
Time in the US (years)	--	--	--	4.2 (5.4)	--
Age at Migration	--	--	--	24 (8.5)	--
Undocumented at Migration	--	--		75	--
Obese	26	17 ^†^	26	29	26
Elevated Waist-Circumference	26	12 ^†^	27	25	26
High Blood Pressure	37	27 ^†^	37	37	36
Elevated Mean Arterial Press.	38	34	38	42	38
Self-Reported Diabetes	6	2 ^†^	6	4	6
Self-Reported CVD	3	3	3	3	3
Current Smoker	14	14	14	18 ^†^	14
Recommended Level of Physical Activity	14	18 ^†^	14	18 ^†^	14

Chi-square tests used to compare categorial variables; *t*-tests used to compare continuous variables; ^†^ Indicates significance at *p* < 0.05 level, * indicates *p* < 0.10.

**Table 2 ijerph-18-12136-t002:** Summary of Mixed-Effects Models of the Association between Cardiometabolic Health and Future Migration to the US, Wave 1, *n* = 14,763.

Dependent Variable:	Future Migrant to the US Migrant = 1; Nonmigrant = 0		
Health Indicator:	OR	s.e.	95% CI
Elevated Waist Circumference	0.61 *	0.16	0.37–1.02
Elevated MAP	0.78	0.15	0.53–1.15
Diabetes	1.05	0.55	0.38–2.95
Cardiovascular Disease	2.20	1.20	0.76–6.38
Smoker	1.35	0.36	0.81–2.27
Physical Activity	1.34	0.33	0.84–2.16

Mixed-effects logistic regression models predicting future migration to the US. Each health indicator was entered alone as an independent variable; controls include age, gender, marital status, education, employment status, insurance status, household assets, urbanicity, smoking status, and physical activity. * Indicates significance at *p* < 0.10. Model adjusted for level 1 clustering at the family level and level 2 clustering at the locality level.

**Table 3 ijerph-18-12136-t003:** Summary of Mixed-Effects Models of the Association between Migration History to the US and Cardiometabolic Health, MxFLS Wave 1, *n* = 14,763.

Migration History Variable	Return Migration (Return Migrant = 1; Stayer = 0)				Time in the US (Years)			
Health Variable:	OR ^1^/b ^2^	s.e.	95% CI		OR ^1^/b ^2^	s.e.	95% CI	
Waist Circumference ^1^	1.49 ^†^	0.67	0.19	2.80	0.27 ^†^	0.10	0.07	0.47
Obese ^2^	1.41 ^†^	0.22	1.03	1.91	1.04	0.02	0.99	1.08
Elevated MAP ^2^	1.08	0.16	0.81	1.44	1.01	0.02	0.97	1.06
High Blood Pressure ^2^	0.98	0.14	0.73	1.30	1.02	0.02	0.98	1.06
Diabetes ^2^	0.71	0.25	0.36	1.43	1.02	0.04	0.94	1.10
Cardiovascular Disease ^2^	1.20	0.42	0.60	2.40	1.06 ^†^	0.03	1.003	1.12

Mixed-effects regression models of the association between return migration and health and between time in the US and health; ^1^ beta coefficients are reported for dependent variables that were modeled using mixed-effects linear regression; ^2^ odds ratios are reported for dependent variables that were modeled using mixed-effects logistic regression. Controls include age, gender, marital status, education, employment status, insurance status, household assets, urbanicity, smoking status, and physical activity. ^†^ Indicates significance at *p* < 0.05 level. Models were adjusted for level 1 clustering at the family level and level 2 clustering at the locality level.

## Data Availability

All codes for data cleaning and analysis associated with the current submission are available from the corresponding author, AMC, upon request.

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
