# Peer review of "The Weight of Migration: Reconsidering Health Selection and Return Migration among Mexicans"

_ijerph, 2021, doi:10.3390/ijerph182212136_

Round 1

Reviewer 1 Report

I do not have particular comments.

See attached document for minor spelling checks.

Author Response

We thank Reviewer 1 for catching several editorial issues and their overall review of our paper. We have edited these errors per your suggestions (see attachment). 

Reviewer 2 Report

I enjoyed reading your well-written manuscript. I am sure that it would interest the many readers of the journal.

The manuscript presents an original research results. The statistics were performed to a high technical standard. the Conclusions are presented in an appropriate fashion and are supported by the data. The article is presented in an intelligible fashion. 

1) Please explain more detail how was the sample selected (size of samples: future migrants, nonmigrants, return migrants, stayers)?

2) What are implications for practice of your research?

Author Response

We thank the Reviewer for their clarifying questions for our manuscript.

Point 1: Please explain more detail how was the sample selected (size of samples: future migrants, nonmigrants, return migrants, stayers)?

Response 1: More detail about the sampling design and sample selection were added in lines 107-111. We also add more information about the migration variables and how the migrant groups were selected and created (lines 121-132). Throughout the manuscript we add the sample size of the migrant groups (e.g. lines 90, 128, 236). 

Point 2: What are implications for practice of your research?

Response 1: In the discussion we add our recommendations for clinical, policy, and practice interventions to ensure the health of Mexican migrants in the US and return migrants in Mexico in light of our findings (line 386). 

Reviewer 3 Report

I would like to sincerely thank the authors for the research carried out on an intertwined subject matter of health and return migration. Both are of outmost interest and relevance in terms of research and policy-setting. It was my pleasure to review a draft of such good quality.

The article looks at their relationship and offers well-supported evidence based conclusions, with recommendations. The draft is well written, the main argument is clearly presented, researched, and discussed.   The evidence-based conclusions offer grounds for recommendations. It complies with standards for accuracy in references and definitions.

Abstract: offers, as it should, a synopsis of the article.

Introduction: is detailed, which in this particular case is a plus, as it sets the scene for the research and explains scientific context as well as migration-specific health conditions for the selected category of migrants in the USA.

Methods: the sample size is representative, and selection criteria for choosing the sample is well justified and described with a good level of detail. research questions are clearly presented, along with the interview method.

Analysis: the tools deployed for the analysis are well described.

Results: here, amongst other topics, return migration represents a particularly relevant matter to explore, given the increasing research and policy interest in return (and reintegration).

Discussion: focuses on the findings confirming the similarities in  health profiles before migration and different health outcomes among return migrants in vis-à-vis stayers. It also acknowledges such hard to measure factors as personal stressors, associated with migration.  

Conclusions benefit from a suggestion for a direction of future work, which should explore social and health environments faced by Mexican migrants to the US. I sincerely hope that the authors will continue their meaningful research work. Another suggestion to consider is to look at reintegration of returnees, from a perspective of health.  

Author Response

We thank the reviewer for their positive review of our paper. While there were no major changes requested, we did include a recommendation that future research should explore how reintegration shapes the health of return migrants, per the reviewers suggestion. 

This manuscript is a resubmission of an earlier submission. The following is a list of the peer review reports and author responses from that submission.